# Root Foraging Capacity in Bambara Groundnut (*Vigna Subterranea* (L.) Verdc.) Core Parental Lines Depends on the Root System Architecture during the Pre-Flowering Stage

**DOI:** 10.3390/plants9050645

**Published:** 2020-05-19

**Authors:** Kumbirai Ivyne Mateva, Hui Hui Chai, Sean Mayes, Festo Massawe

**Affiliations:** 1Future Food Beacon Malaysia, School of Biosciences, University of Nottingham Malaysia, Jalan Broga, Semenyih 43500, Selangor Darul Ehsan, Malaysia; hbxkm1@nottingham.edu.my (K.I.M.); huihui.chai@nottingham.edu.my (H.H.C.); 2Crops For the Future, Jalan Broga, Semenyih 43500, Selangor Darul Ehsan, Malaysia; sean.mayes@nottingham.ac.uk; 3School of Biosciences, University of Nottingham, Sutton Bonington Campus, Loughborough, Leicester LE12 5RD, UK

**Keywords:** bambara groundnut, branching, deep rooting, drought adaptation, root traits

## Abstract

Characterizing the morphological variability in root system architecture (RSA) during the sensitive pre-flowering growth stage is important for crop performance. To assess this variation, eight bambara groundnut single genotypes derived from landraces of contrasting geographic origin were selected for root system architecture and rooting distribution studies. Plants were grown in a polyvinyl chloride (PVC) column system under controlled water and nutrient availability in a rainout shelter. Days to 50% plant emergence was characterized during the first two weeks after sowing, while taproot length (TRL), root length (RL), root length density (RLD), branching number (BN), branching density (BD) and intensity (BI), surface area (SA), root volume (RV), root diameter (RDia), root dry weight (RDW), shoot dry weight (SDW), and shoot height (SH) were determined at the end of the experiment, i.e., 35 days after emergence. Genotypes S19-3 and DipC1 sourced from drier regions of sub-Saharan Africa generally had longer taproots and greater root length distribution in deeper (60 to 90 cm) soil depths. In contrast, bambara groundnut genotypes from wetter regions (i.e., Gresik, Lunt, and IITA-686) in Southeast Asia and West Africa exhibited relatively shallow and highly branched root growth closer to the soil surface. Genotypes at the pre-flowering growth stage showed differential root foraging patterns and branching habits with two extremes, i.e., deep-cheap rooting in the genotypes sourced from dry regions and a shallow-costly rooting system in genotypes adapted to higher rainfall areas with shallow soils. We propose specific bambara groundnut genotype as donors in root trait driven breeding programs to improve water capture and use efficiency.

## 1. Introduction

Root system architecture (RSA) describes the form and spatial structure within the soil of a root system [1]. This has significant implications for plant development and enables plants species to adapt to environmental cues in order to flourish in various ecological habitats [2]. Variations in RSA are related to differences in soil nutrient and water acquisition among landraces of a similar developmental form but originating from contrasted ecological niches [3]. Bambara groundnut (*Vigna subterranea* (L) Verdc), is an exemplar neglected African grain legume that thrives under strikingly contrasted environments relative to other grain legumes. Originating in West Africa, its distribution spans across climatic gradients from Senegal to Kenya and from the Sahara to South Africa with recent introductions in Southeast Asia [4]. In these contrasting habitats, bambara groundnut has diversified due to domestication from its wild relative, Vigna subterranea var. spontanea (Harms) Hepper, as a result of steady changes through natural and artificial selection [5].

Looking at the different patterns of root distribution, theoretical and experimental research propose that a root system comprising of a deep-cheap rooting system associated with a long taproot system and few primary first-order laterals, would favor deep water foraging and mobile nutrients acquisition in low-resource habitats [6]. Conversely, a shallow-costly rooting system associated with a shorter taproot system and greater primary first-order laterals, would favour water and nutrient acquisition in the shallow soil depths of high and low-resource habitats [6]. Generally, leguminous crop species cultivated in hot-dry environments exhibit a particular root topology, i.e., long taproot system with few primary first-order laterals [7]. However, detailed descriptions are currently missing for root trait differences among bambara groundnut genotypes of contrasted habitats.

Considering the increasing shortage of agricultural water and that no single shoot trait has yet been identified for its unique and dominant contribution to drought resistance [8,9,10,11,12,13,14,15,16,17,18,19], current bambara groundnut breeding efforts could investigate root system function and its manipulation in order to improve water and nutrient capture [20,21,22]. Bambara groundnut genotypes, such as S19-3 (from Namibia) are promising candidates for investigating and expanding plant ideotypes suited for dry environments [11,23]. In reality, for many centuries, southern African farmers have selected local bambara groundnut genotypes for their drought resistance [24]. As it occurred in other crop species [25,26,27,28], indirect selection by farmers for improved rooting capacity is likely to have occurred in bambara groundnut through its influence on yield over the years. Investigating the morphological variability in RSA and primarily the contrast between a diverse collection of bambara groundnut genotypes sourced from various agroecologies, would help test the hypothesis of indirect selection, while also defining root system ideotypes that could improve soil resource uptake.

Since grain legumes suffer significant yield reduction due to water deficit stress that occurs earlier on during the reproductive growth stages [29], characterizing the root system at final harvest cannot reveal the range of maximum variation for drought resistance breeding. Significant variation in root growth is observable just before flowering, i.e., 35 to 45 days after emergence (DAE), [30,31,32,33,34]. As such, initial characteristics in plant root development and branching manner are essential to the ultimate establishment of the plants and their subsequent exploration of the soil volume for water. Therefore, root trait variation observed at this stage would aid in determining the most informative root traits that significantly reduce grain yield penalties for subsequent breeding [35].

From this, an initial exploration of genetic variability in root characteristics in bambara groundnut would be an important step to developing varieties for target environments. This would be important in complimenting and possibly explaining recent bambara groundnut findings limited to only shoot morpho-physiological traits. By means of a polyvinyl chloride (PVC) column study, the present experiment explores the developmental variation in taproot and branching patterns at the pre-flowering stage using a collection of single genotypes derived from landraces of contrasting agroecological backgrounds. The underlying speculation is that bambara groundnut plants from low resource agroecologies have throughout the years developed root traits that improve resource foraging in deep soil depths. All the more explicitly, it is expected that bambara groundnut plants from low resource habitats would have more extensive root extension in the deep soil depths during the early growth stage. 

## 2. Results

### 2.1. Plant Emergence, Size, Biomass Production, and Root/Shoot Ratio

According to results of analysis of variance, seed emergence, shoot height, and root/shoot ratio were significantly affected by genotype (Table 1). Seed emergence started three days after sowing (Gresik) and continued up to 10 days (Ankpa-4) with a mean of six days. The genotype Gresik showed the fastest emergence, resulting in higher shoot height and biomass production at 35 DAE, as well as higher number of leaves, although this was not statistically different from the other genotypes. Days to 50% emergence was negatively and highly correlated to the root dry weight (*r* = −0.39, *p* < 0.006) and, subsequently, the root to shoot ratio (*r* = −0.47, *p* < 0.001) (Appendix A), demonstrated a root biomass decline as a result of slow seedling emergence. At that time, Ankpa-4, the least vigorous of the studied genotypes, was only 78% the size of Gresik in terms of shoot height and 34% its size in terms of root dry weight. Lunt, IITA-686, Dodr, S19-3, Tiga nicuru, and DipC1 showed intermediate and statistically similar values for emergence, with Tiga nicuru less productive (for root dry weight, 0.28 g plant^−1^) than Gresik (0.64 g plant^−1^, Figure 1), essentially due to lower shoot dry weight (1.19 g plant^−1^). Despite contrasted growth capacities, the genotypes Gresik and S19-3 showed similar biomass allocation patterns with root/shoot ratios of 0.35 and 0.28, respectively, significantly higher than those observed in Ankpa-4 (0.16, Table 1).

### 2.2. Deep Rooting Profile

After 35 DAE, the vertical growth of the root system showed significant differences among genotypes (Figure 2A,B). The genotypes varied significantly for the rooting depth, i.e., taproot length. This ranged from 58.9 cm (Gresik) to 100.6 cm (DipC1) with average taproot length of 78.6 cm (Figure 2B). Compared to DipC1, the genotypes Tiga nicuru and Gresik showed significantly less deep rooting, although the former still penetrated deep soil, i.e., >60 cm while the latter was almost exclusively limited to the 30 to 60 cm layer (Figure 2B). The genotypes S19-3 and Ankpa-4 showed significantly higher root depth, recording the second largest taproot length (95.1 cm and 89 cm, respectively), although S19-3 was not statistically different (*p* > 0.05) from the deepest rooting genotype DipC1 (Figure 2B). Plants of DipC1 had substantially longer taproots, with up to 19-fold more taproot length in the 60 to 90 cm depth as compared with the shallowest genotype (Gresik, 1.36 cm) (Figure 2C).

### 2.3. Root System Branching and Density Dynamics

The bambara groundnut root systems at 35 DAE were limited to the taproot, first-order, and second-order lateral branching (Figure 3A). The analysis of branching numbers and branching density, i.e., first-order lateral roots, from the taproot, revealed contrasted dynamics among the studied genotypes. Total branching numbers ranged from 120 (Tiga nicuru) to 278 (Gresik) with an average of 209 (Figure 3B), with the largest variance in the shallow soil depth (0 to 30 cm) (Figure 3C). Branching density followed a somewhat similar trend, ranging from 1.7 cm^−1^ (Ankpa-4) to 4.5 cm^−1^ (Gresik, Figure 4A,B). The genotypes, Gresik had the most BD (Figure 4B), although not statistically different from IITA-686, (3.6 cm^−1^ taproot length) and in the case of Gresik, this was largely distributed in the shallow soil depth (0 to 30 cm), as shown by data from the branching intensity (Figure 4C). As a result, root length and root length density, direct components of first- and second-order lateral branching were highest in the genotype Lunt (4603.5 cm and 0.13 cm root cm^3^ soil, respectively) followed by the second highest genotype Gresik (4545.5 cm and 0.13 cm root cm^3^ soil, respectively, Figure 5A and Figure 6A).

A more detailed look into the different soil depth segments revealed that branching number in the 0 to 30 cm depth ranged from 75 (Tiga nicuru) to 207 (Gresik, Figure 3C). The least branching genotype, Tiga nicuru, was statistically similar to S19-3 and DipC1 (117 and 116, respectively, Figure 3C). It appeared that changes in branching numbers reflected changes in root length in the 0 to 30 cm, 30 to 60 cm soil depths (*r* = 0.64, *p* < 0.001 and *r* = 0.31, *p* < 0.03, respectively, Appendix A) and not in the 60 to 90cm soil depth (*r* = −0.19, *p* > 0.19), a direct result of low mean BI values (0.000276 cm^−1^ root length) realized in that soil depth (Figure 4C). The genotypes Lunt and Gresik ranked highest for root length (3700.2 and 3430.6 cm, respectively, Figure 5A) and root length density (0.39 and 0.36 cm root cm^3^ soil, respectively, Figure 6A), whilst Tiga nicuru and Ankpa-4 allocated the least. However, S19-3 and DipC1 (genotypes originating from drier regions), branching number and subsequently branching intensity was highest at 60 to 90 cm as soon as 35 DAS, when most of the other genotypes had yet reached that depth. In the 60 to 90 cm soil depth, S19-3 and DipC1 had up to 55- and 34-fold, respectively, longer root length density (0.07 cm root cm^3^ soil) as compared with Gresik (0.001 cm root cm^3^ soil, Figure 6C).

In the shallow 0 to 30 cm soil depth, shoot dry weight was closely and positively correlated (*p* < 0.05) with a wide range of traits, including root length density (*r* = 0.73), root length (*r* = 0.73), root surface area (*r* = 0.74), and root volume (*r* = 0.73, Appendix A). Additionally, shoot height was closely and positively correlated (*p* < 0.05) with branching number in the deep 60 to 90 cm of the soil (*R*^2^ = 0.53), and this was largely amongst genotypes originating from drier versus wetter environments (Appendix A).

### 2.4. Root Surface Area, Volume, and Diameter

Total root surface area ranged from 422.7 to 793.9 cm^2^ (for genotypes Tiga nicuru and Gresik, respectively, Figure 7A). Surface area in the topsoil (0 to 30 cm) depth ranged from 275.1 cm^2^ (Tiga nicuru) to 645.1 cm^2^ (Lunt) with average surface area of 434.9 cm^2^ (Figure 7B). The genotypes, Gresik had the second largest surface area (630.3 cm^2^) in the 0 to 30 cm soil depth segment, although Lunt was not statistically different (*p* > 0.05) from Gresik. However, in deeper soil depths (60 to 90 cm) the genotype DipC1 had substantially more surface area (74.7 cm^2^), with up to 35-fold more surface area as compared with the least (Gresik, 2.1 cm^2^, Figure 7C). Total root volume ranged from 4.31 to 9.78 cm^3^ (for genotypes Ankpa-4 and Gresik, respectively, Figure 7D). Root volume in the 0 to 30 cm topsoil depth varied among genotypes (Figure 7E). Ranging from 3 cm^3^ (Ankpa-4) to 8.3 cm^3^ (Gresik) with an average of 5.2 cm^3^. Although genotype Gresik had the largest root volume (8.3 cm^3^) it was not statistically different (*p* > 0.05) from Lunt (Figure 7E). The genotypes Lunt and Gresik ranked highest for root volume in the 0 to 30 cm soil depth segment (86%, and 85%, respectively, Figure 7F). While DipC1 and S19-3 allocated the least root volume in the same topsoil segment (53% and 58%, respectively). Total root diameter ranged from 1.07 to 1.83 mm (Gresik and S19-3, respectively, Figure 7G). Root diameter in the 60 to 90 cm subsoil depth varied among genotypes and ranged from 0.05 mm (Gresik) to 0.52 mm (S19-3) with an average of 0.38 mm (Figure 7H). The genotypes Gresik and Lunt ranked highest for root diameter in the 0 to 30 cm soil depth segment (49% and 41%, respectively, Figure 7I). While S19-3, DipC1 and Ankpa-4 allocated the least root volume in the topsoil segment (25%, 28%, and 29%, respectively) and more in the deeper soil (20%, 18%, and 12%).

### 2.5. Identification of Grouping of Genotypes with Relatively Homogeneous Root Traits

In the bambara groundnut parental line collection, the number of genotypes included from Southeast Asia was (*n* = 1), West Africa (*n* = 3), East Africa (*n* = 2), and from Southern Africa (*n* = 2). Four relatively homogeneous genotype groups were determined based on a K-means clustering analysis (Appendix A). This indicated that high contrasting genotypes for the root traits studied could be distinguished and confirmed from the soil-filled PVC column system.

The outlier genotype, Gresik (Cluster 1), from Southeast Asia, was separated from the others and recorded a significantly shorter taproot (Figure 8A) with the highest ranked RLD in the 0 to 30 cm topsoil depth (Figure 8B). Cluster 2 contained three genotypes, i.e., Lunt (west Africa), IITA-686, and Dodr (both East Africa) representing an intermediate taproot and RLD in the topsoil 0 to 30 cm depth. Similarly, Cluster 3 had two genotypes (Tiga nicuru and Ankpa-4 both from West Africa) also representing an intermediate taproot system and RLD in the same soil depth. In contrast, genotypes that originated from Southern Africa, i.e., DipC1 and S19-3 (both in Cluster 4), had significantly deeper taproot systems than the other three regions (Figure 8A) with also significantly less RLD in the shallow soil depth (Figure 8B).

## 3. Discussion

In the last 30 years, drought breeding work on bambara groundnut has been limited and largely focused and elucidated by above ground shoot morpho-physiological studies [11,13,14,16,19,36,37,38,39,40,41,42]. In order to fully understand and better manipulate the ability to tolerate water-limited conditions, the range of both above- and belowground variation present in bambara groundnut germplasm needs to be explored [43]. Unlike the former, belowground plant root research has been mainly hampered by the difficulty to access the rhizosphere [44]. Considering that root traits influence water acquisition and, subsequently, yield [32], it is anticipated that the variability in the root traits of different bambara groundnut genotypes reported in this paper could be the missing link and initial step toward finally understanding bambara groundnut’s superior drought adaptation.

Various platforms have been proposed to study the root system, most of which do not allow plant research and root analysis in a soil substrate. By modifying an efficient low-cost soil-filled PVC column phenotyping system [45,46], quantitative comparisons for root traits among different bambara groundnut genotypes were made possible. Although not high throughput, the screening system allowed for natural soil and physical properties such as bulk density to be mimicked. Using this system, previously unknown root variation in a contrasting collection of bambara groundnut genotypes at 35 days after emergence (DAE) was determined. 35 DAE represents the mean pre-flowering stage of bambara groundnut, a growth stage sensitive to drought stress with strong subsequent effects on yield and yield parameters [8,23]. Studies by [47,48], found that genotypes with larger early and high pre-flowering root length densities penetrate deeper soil layers which improves drought resistance with significantly higher grain yield under early water deficit conditions. In addition, wide genotypic variation has been reported at the pre-flowering growth stage in a number of cereal and legume root-related studies and our current work on bambara groundnut is no exception. Studies on chickpea, lupin (*Lupinus angustifolius*), common bean (*Phaseolus vulgaris*), and wheat (*Triticum aestivum*) indicate that significant variation in root growth is observable from 35 to 45 DAE [30,31,32,33,34]. Therefore, root trait variation observed at this stage would aid in determining the most informative root traits that confer grain yield advantage during terminal drought stress.

The genotype Gresik had the earliest emergence, i.e., 4.8 days after sowing (DAS), however DipC1 plants emerged 2.4 days later (Table 1) still managing to penetrate deep soil layers faster than the wet region genotypes. Even so, days to 50% emergence was found to be correlated to the total taproot length (*r* = 0.44, *p* < 0.001, Appendix A), and this particular case confirms DipC1′s elongation dynamics associated with a quick and deep rooting system, an adaptation mechanism to dry environmental conditions.

Bambara groundnut root systems, as with many dicotyledons, are characterized by a well-defined taproot system, with numerous first-order lateral branches. These lateral roots further branch into second- and third-order laterals. While the topology, i.e., taproot and primary laterals, remained virtually similar among the eight bambara groundnut genotypes, differences in rooting depth and branching were observed in different soil depth segments at 35 DAE. The genotype Lunt produced higher total root length in the first 0–30 cm of the soil than Gresik (the outlier genotype, see Appendix A) and all the other studied genotypes, though the difference between the Lunt and Gresik appeared marginal. In addition, the genotype Dodr, Lunt and IITA-686 were found to have similar extensive lateral branching (comprising of large root surface area, volume and diameter). A more detailed analysis of root length revealed that root branching density and intensity were not only higher in the topsoil layer: they were also more abundant in genotypes originating from the wetter regions that experience sporadic rainfall throughout the growing season, than in the dry region genotypes. Conversely, Gresik had statistically shorter taproot length but consistently high root branching density and intensity values, bringing about higher total root length and root length densities in the shallow soil layer 0 to 30 cm at the end of the 35 DAE. Such compensation between taproot length and branching, bares a general ecological significance as revealed by [49] who found that by studying independent contrasts among Australian perennial plants, species originating from wet habitats typically have high root proliferation in the shallow soil depth as compared with species from dry habitats. In humid climates such as Indonesia from where Gresik was collected, rainfall wets the soils frequently. With frequent topsoil wetting the genotype Gresik’s roots do not need to forage for deep water reserves. From a functional perspective, a costly highly branching system in the shallow topsoil layer improves the root absorption of phosphorous, however, in the case of a drought, this would enhance water depletion in that layer because of acute root competition [50].

In the most arid parts of Southern African, bambara groundnut is often grown after or intercropped with major cereal crops such as sorghum (*Sorghum bicolor*) and maize (*Zea mays*) [51], and this is towards the end of the main wet season. The crop is forced to survive on residual soil moisture exposing the crop to terminal drought stress. In such cases, the crop has to quickly establish and develop a deep rooting system with an optimal lateral root length investment in energy to maximize capture of stored soil moisture at depth more efficiently. This attribute was indeed observed in genotypes from dry regions (DipC1 and S19-3) which produced limited lateral roots in the shallow soil depths but had long deep taproots > 90 cm depth at 35 DAE, when none of the other genotypes had reached that depth yet (Figure 2B). At the pre-flowering growth stage, plant roots that are able to reach deeper soil depths would support flowering with improved yield formation under drought [52]. Differences in root architecture among genotypes from hot-dry and humid-wet regions suggest an adaptive response of bambara groundnut for soil resource capture by means of an improved foraging capacity of the root system in the hot-dry region sourced genotypes [53,54]. Interestingly, Gresik was derived from an introduction into Indonesia from Africa not so long ago and most likely from Southern and East Africa [55]. Practically, it would make sense that Gresik would need to have adapted quickly into a costlier highly branching root system in the shallow topsoil layer, given the conditions in Indonesia. In such a case, environmentally responsive genes could have played a role, leading to root plasticity in order to enhance its growth in wetter environments. A study by Jørgensen et al. [11], defined S19-3 as a ”water-spender” exhibiting late closure of stomata and, consequently a slow decline in transpiration rate during drought. Accordingly, this mechanism is now best supported by our root findings and classification of S19-3 as a genotype with an extensive root length density in the deeper soil layers as compared with the topsoil layer. The importance of increasing root and soil contact through greater root length density is that it allows plants to access greater quantity of soil [56,57] in water-limited environments. Such positive correlations have been observed between deep root systems and drought resistance of chickpea, common bean, sugar beet (Beta vulgaris), and maize [58,59,60]. Similarly, [61] demonstrated that an increase of 30 cm rooting depth allows for the capture of an extra 10 mm of water in the deeper soil layers, resulting in an additional (0.5 t ha^−1^) of wheat grain. Allowing the crop access to deep water reserves long after a drought event has started. Therefore, early selection (i.e., 35 DAE) for greater root length density at depth can be expected to help enhance the genetic gains and yield improvement in bambara groundnut breeding efforts. Such a root system reduces the metabolic costs that comes with having to maintain an elaborate root architecture, thus, allocating more resources towards deep soil foraging in order to access deep water and mobile nitrogen [62]. Consequently, because S19-3, much like DipC1, originated from drier regions (Namibia and Botswana, respectively), where increased vapor pressure deficit (VPD) increases atmospheric demand for transpired water. It is tempting to speculate that DipC1, with the genetic predisposition for long taproot system (Figure 2B), could also be coined a ”water spender” and could adopt similar physiological mechanisms such as the ones of its documented counterpart. Further work is needed to test this hypothesis.

The differences in root systems observed between the four geographic location, reflect contrasting strategies for adaptation to environments with different rainfall patterns. This result, along with previous bambara groundnut shoot phenotyping, presents proof that root trait variability in bambara groundnut is as important as shoot trait phenotyping and contributes to plant survival and yield under water limited conditions. Given that the root system is a hidden and complex organ, the prospects of indirect selection by utilizing aboveground plant parts becomes highly desirable. Shoot dry weight was positively associated with root length density, root length, root surface area, and root volume all in the shallow soil depth. In addition, shoot height was positively correlated with the number of branches in the deep 60 to 90 cm of the soil. This indicates that shoot dry weight and shoot height are good traits that can be used as proxies to make estimations of several shallow and deep root traits, respectively, in bambara groundnut and could both be prioritized for large-scale breeding phenotyping. As such, we conclude that farmers over the years have indirectly selected for differences in deep rooting and root length density through their influence on yield under dry environments [27,45,54]. Furthermore, bambara groundnut genotypes that evolved in drier areas could have adapted by increasing taproot length and reducing their branching distribution to capture deep water more efficiently. However, these traits cannot be of any advantage in humid environments with high annual average rainfall [46] where a short rooting and highly branched system in the superficial soil layers would seem less adapted to drought.

## 4. Materials and Methods

### 4.1. Plant Material

A collection of eight bambara groundnut single genotypes derived from landraces of contrasting geographic origin were selected for the root trait variability studies (Table 2). In detail, the seeds were collected from seven countries and four geographical regions, i.e., West Africa (*n* = 3), East Africa (*n* = 2), Southern Africa (*n* = 2), and Southeast Asia (*n* = 1, Table 2), with most of the landrace names based on the place the seeds were collected [63]. These genotypes are representatives of the bambara groundnut parental line collection currently being screened for drought resistance as part of a current ongoing project by Crops For the Future (CFF) [64].

### 4.2. Study Site and PVC Column Screening System

Polyvinyl chloride (PVC) column experiments were conducted under a rainout shelter during two consecutive seasons (2017–2018 and 2018–2019) at the Crops For the Future-Field Research Center (CFF-FRC) located at 2°55’52.2”N 101°52’45.7”E, altitude 42 m above sea level in Semenyih, Malaysia. The bambara groundnut core parental lines were grown in light weight PVC pipes of 20 × 116 cm (inside diameter and length, respectively, Figure 9A) under rainout shelter in a completely randomized (CRD) design in six replications for each year.

The pipes were placed on top of a detachable perforated plate which allowed free drainage of excess water. In addition, a wooden frame was constructed to support the columns and keep them upright (Figure 9B). To facilitate root harvesting, each column was cut longitudinally along both sides and the two halves taped together with 4.8 cm (by width) packing brown tape before filling with soil. Soil used in the experiments was composed of a mixture of air-dried sand and clay (2:1 *w*:*w*). The soil was sieved through a 3 mm mesh to eliminate ˃ 3 mm diameter soil particles. The soil was poured into the PVC pipe and manually packed using an ~1.5 kg concrete base tamper, with base plate diameter equal to the internal diameter of the pipe. The soil surface was lightly scrapped after every interval pack to provide hydraulic connectivity between the soil portions, preventing a layering effect [65]. Each pipe was packed to allow for a homogenous continuum, rather than stratified layers. The downward movement of water and growth of roots is limited by horizontally stratified layers. As such, soil packing allowed all soil fractions to be exposed equally to water, promoting uniform water distribution rather than preferential flow pathways [66]. Pipes were filled with 55.3 kg of soil up to 110 cm high achieving a constant bulk density of 1.6 g cm^−3^ and hereafter referred to as columns. Basal fertilizer (10 kg·ha^−1^ N as urea (46%), 50 kg·ha^−1^ P as Christmas island rock phosphate (CIRP) (30%), and 50 kg·ha^−1^ K as muriate of potash (MOP)(60%) was surface applied and incorporated into the topsoil layer (0 to 10 cm) of the column. In addition, 90 g of granular Agromate ABC micronutrients (Agromate International, Ltd) were dissolved in 10 L of water and added to the soil columns at two and three weeks after emergence (WAE). The solution consisted of Mn EDTA (3.8%), Fe EDTA (4.0%), Cu EDTA (1.5%), B (0.5%), Zn EDTA (1.5%), Co (0.03%), Mo (0.10%), and Mg (5.10%).

After removing broken and damaged seeds, uniform sized seeds were selected and surface sterilized in a 10% (*v*/*v*) Clorox solution (sodium hypochlorite 0.5%) for 2 min on a rotary shaker at 150× *rpm*. Following this, seeds were rinsed thrice using distilled water. Sterilized seeds were placed in 9 cm diameter petri dishes and allowed to imbibe water for 15 h at a temperature of 28 ± 1 °C, in the dark. For each genotype, two seeds were sown in individual columns. One healthy representative plant per column was maintained and the plants were grown under field conditions and protected from rainfall using a fixed-location transparent acrylic rainout shelter. The columns were irrigated until seedlings emergence (an average of 6 days to emergence across the studied genotypes) and, then, irrigated with 250 mL of water four times on alternate days until harvest, i.e., 35-days after emergence (DAE). Two insecticides, i.e., Agus 24SC at a rate of 16 mL 10 L^−1^ of water and Akosu 9.5SC at a rate of 7.5 mL·10 L^−1^ of water (active ingredient, diafenthiuron 24.0% and chlorfenapyr 9.5% both suspension concentrates) were prepared as a tank mix and sprayed every seven days to protect the plants from white flies (*Aleyrodidae* spp.) and red spider mites (*Tetranychus* spp.), respectively. Fungicide with active ingredient: didenoconazole 20.0% emulsifiable concentrate was sprayed once at three WAE at a rate of 10 mL 10 L^−1^ of water.

### 4.3. Plant Sampling and Measurements

To extract the roots, the PVC column was laid down and tilted at a 20° angle to the root washing station. The detachable perforated PVC plate was removed, and the column split in half longitudinally (Figure 9C). The soil was gradually removed to expose the roots in a bottom-up manner using soft spray watering head. After complete removal of the soil, the shoots (i.e., leaves and stems) were separated from roots and entire root systems were submerged in water-filled zip lock bags of 22 × 30 cm (width and length, respectively) and transported to the laboratory for further assessment.

In order to identify and measure the taproot length (TRL), entire roots were laid flat and stretched against a two-meter ruler, giving an estimate of the deepest extent of the root system. For the purposes of this paper, entire root systems (i.e., totals) were analyzed first. Following totals analysis, root systems were cut into different segments with respect to varying 30 cm soil depth (i.e., 0 to 30, 30 to 60, 60 to 90, and 90 to 110 cm) and analyzed as such. In both cases, roots were spread in a shallow A3 size, 297 × 420 mm (height × width, respectively) clear acrylic tray filled with water and disentangled using plastic forceps to reduce overlapping. Root traits were all computed from the scanned images in greyscale at 400 dots per inch using a flatbed Epson Scanner (Epson Perfection V700, CA, USA) with WinRhizo Pro software v2009 (Regent Instruments, Montreal, QC, Canada). These included root length (RL cm), representing root lengths in the network. Branching number (BN), the number of first-order lateral roots emerged from the taproot. Root surface area (SA cm^2^), root volume (RV cm^3^) and root diameter (RDia mm) assessed as proportionate estimations of RL and expected to exhibit the same pattern and trend of variation [45]. These traits subsequently allowed for the calculation of root length density (RLD cm cm^3^), branching density (BD), and branching intensity (BI) using the following formulae:Root length density (RLD) = root length (cm)/soil volume (cm^−3^),(1)
Branch density (BD) = number of branches/taproot length,(2)
Branch intensity (BI) = number of branches/root length depth segment^−1^.(3)

Shoot height (SH) was recorded on a fresh plant basis from the root crown to the apex of the longest plant stem using a ruler. To measure biomass accumulation, shoot dry weight (SDW) and root dry weight (RDW) were recorded after drying in an oven at 80 °C for 72 h and expressed as g plant^−1^.

### 4.4. Developmental Traits

Days to 50% emergence (D50%) was recorded as the number of days after planting when 50% of the plants per genotype had emerged from the soil.

### 4.5. Data Analysis

General linear model (GLM) multivariate analysis was performed for genotypes as main effects using Statistica Version 13.3 software (TIBCO Inc, USA). The significance of the main effect of the season was assessed using the [67] statistic that asymptotically follows a χ^2^ distribution. Wald statistics revealed that the error components across the years (G × Y) for the traits, were homogenous, and therefore it was necessary to draw inferences combined across years for the measured traits. Non-normally distributed data were transformed before analysis of variance (ANOVA) using the square root transformation, for the following traits: taproot length (TRL), root length (RL), surface area (SA), root diameter (RDia), root volume (RV) (all 90 to 110 cm depth), branching density (BD) 60 to 90 cm, branching intensity (BI) 30 to 60 cm, and 60 to 90 cm depths. Mean comparisons were performed on the transformed scales and compared using post-hoc Tukey’s honest significant difference (HSD) at significance level of 95%. For the presentation of the results, the means were back-transformed. Correlations between traits was performed using the cor ( ) and corrplot ( ) functions from the corrplot package in R (R Core Team, 2017). Since the traits have different units, they were scaled to have a variance of one and a mean of zero, using the Standardize function in Statistica Version 13.3. The eight bambara groundnut genotypes exhibited distinctly variable morphologies, and therefore reduced the complexity of the data, K-means clustering was used to generate homogeneous clusters of genotypes. K-means clustering was run with different numbers of clusters (Clusters 3 to 4). Four clusters provided the most interpretable output, in terms of genotype clustering by traits.

## 5. Conclusions

The present study is an initiative to better understand an ignored African grain legume with superior drought resistance relative to other cultivated grain legumes in Africa. To the best of our knowledge, we provide the first itemized report of RSA in core bambara groundnut parental lines. In general, the deep taproot systems and fewer first-order lateral root branching conferring an efficient soil exploration, make a suite of root traits that have over the years fundamentally improved the water foraging capacity of S19-3 and DipC1 as compared with Gresik, Lunt, IITA-686, and Dodr. This is especially valid for hot-dry-habitat S19-3 and DipC1, which flourish in an area of deep sandy soils under very dry and hot climate and, in the case of our study, demonstrated the most noteworthy rooting traits. With respect to the outlier genotype Gresik, it showed a particular root growth pattern best suited to shallow soils that receives frequent wetting. In the two circumstances, specific sets of RSA are expressed from the pre-flowering growth stage to support initial plant establishment. The distinctly differentiated root morphologies concur with two differential foraging strategies in dry environments, namely shallow-costly root systems exploring topsoil layers to beneficiate from occasional rainfall, versus deep-cheap root systems foraging water stored in deeper soil depths [3]. Bambara groundnut root trait study could be exploited in breeding for enhanced drought adaptation or low-input farming, though this would require some correlative investigations to confirm whether pre-flowering root traits translate into improved performance of mature plants in the field [68,69]. More so, the genotypes originating from southern and eastern African region possessed the best deep rooting, indicating potential for further selection from these regions. From an evolutionary point of view, it is important to note that crop domestication and natural selection depend on phenotypic selection [21]. Considering that the larger shoot height is related to more branching in the deep soil layers of dry region genotypes and often results in higher yield than in its wet region sourced bambara groundnut genotypes counterpart. We hypothesize that farmers, over the years, have indirectly selected for differences in deep rooting and root length density, in particular through their influence on yield under dry environments. Moving forward, instead of adopting a strict “back to the roots” framework, bambara groundnut root phenotyping could prioritize these specific root traits.

## Figures and Tables

**Figure 1 plants-09-00645-f001:**
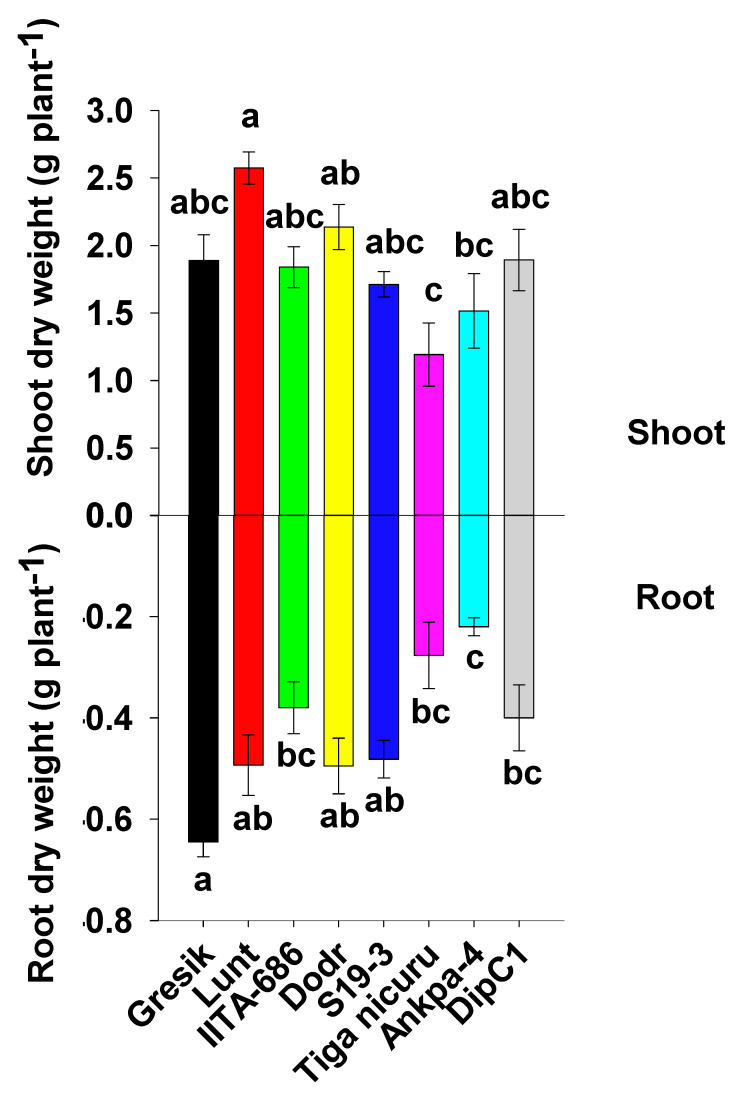
Shoot dry weight (SDW) and root dry weight (RDW) for bambara groundnut genotypes at 35 days after emergence (DAE) grown in a soil-filled PVC column of 20 × 110 cm (diameter and length, respectively). Mean ± se values (*n* = 12) are shown. Different letters indicate significant differences (HSD, *p* < 0.01).

**Figure 2 plants-09-00645-f002:**
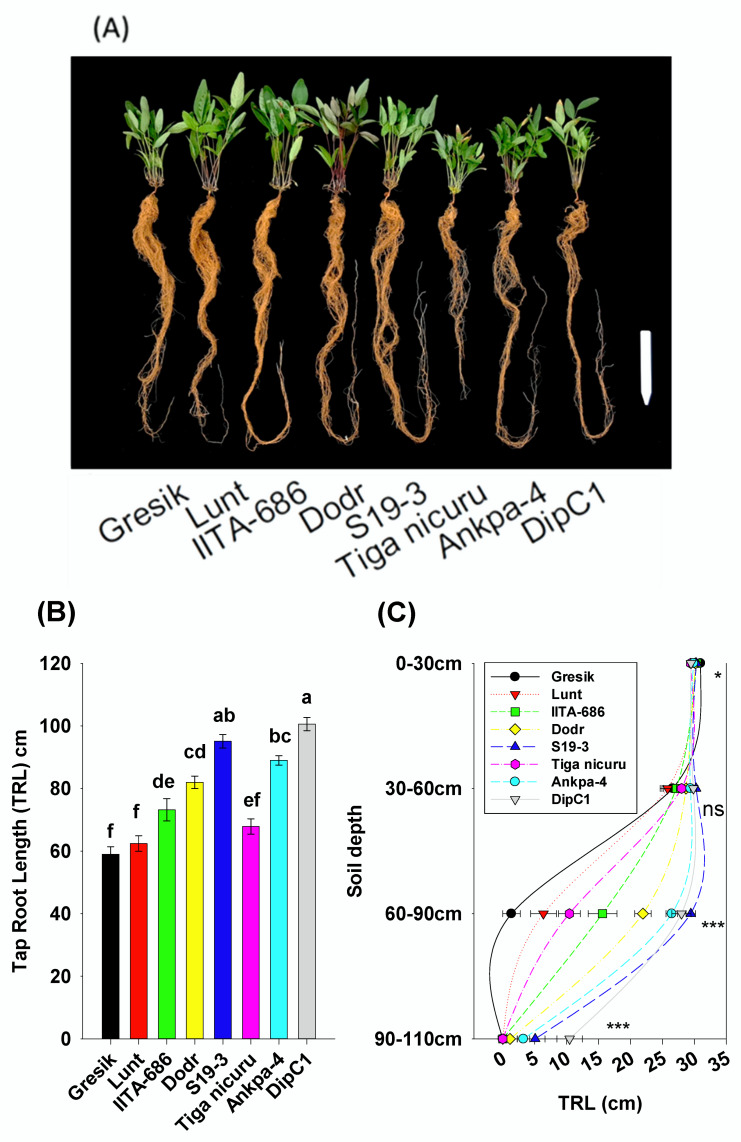
(**A**) Images of the entire root system for bambara groundnut genotypes at 35 days after emergence (DAE) grown in a soil-filled PVC column of 20 × 110 cm (diameter and length, respectively). White bar = 15 cm; (**B**) Total taproot length (TRL) in bambara groundnut genotypes. Mean ± se values (*n* = 12) are shown. Different letters indicate significant differences (HSD, *p* < 0.01); (**C**) TRL’s per soil depth segments. Mean ± se values (*n* = 12) are shown. Significant differences as follows: * *p* < 0.05; ** *p* < 0.01; *** *p* < 0.001; and ns = not significant, amongst individual genotypes.

**Figure 3 plants-09-00645-f003:**
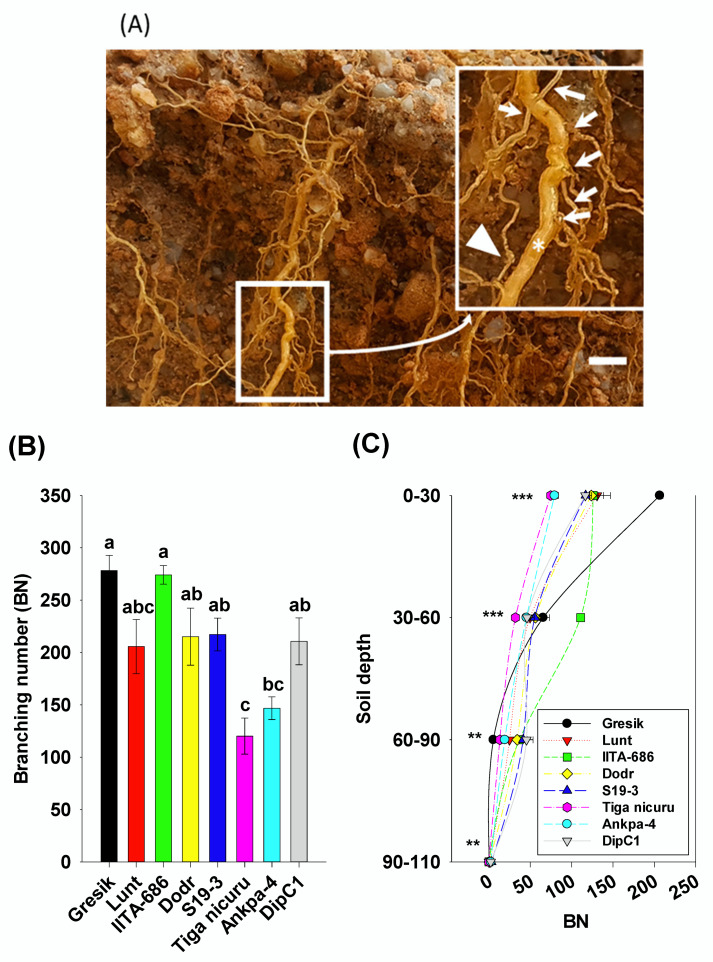
(**A**) An enlargement of the bambara groundnut taproot (asterisk), first-order laterals (arrows) and second-order lateral roots (arrowhead) for the genotype Gresik at 35 days after emergence (DAE) grown in a soil-filled PVC column of 20 × 110 cm (diameter and length, respectively). White bar = 0.5 mm; (**B**) Total branching number (BN) of first-order lateral roots. Mean ± se values (*n* = 6) are shown. Different letters indicate significant differences (HSD, *p* < 0.01); (**C**) BN’s per soil depth segments. Mean ± se values (*n* = 6) are shown. Significant differences as follows: * *p* < 0.05; ** *p* < 0.01; *** *p* < 0.001; and ns = not significant, amongst individual genotypes.

**Figure 4 plants-09-00645-f004:**
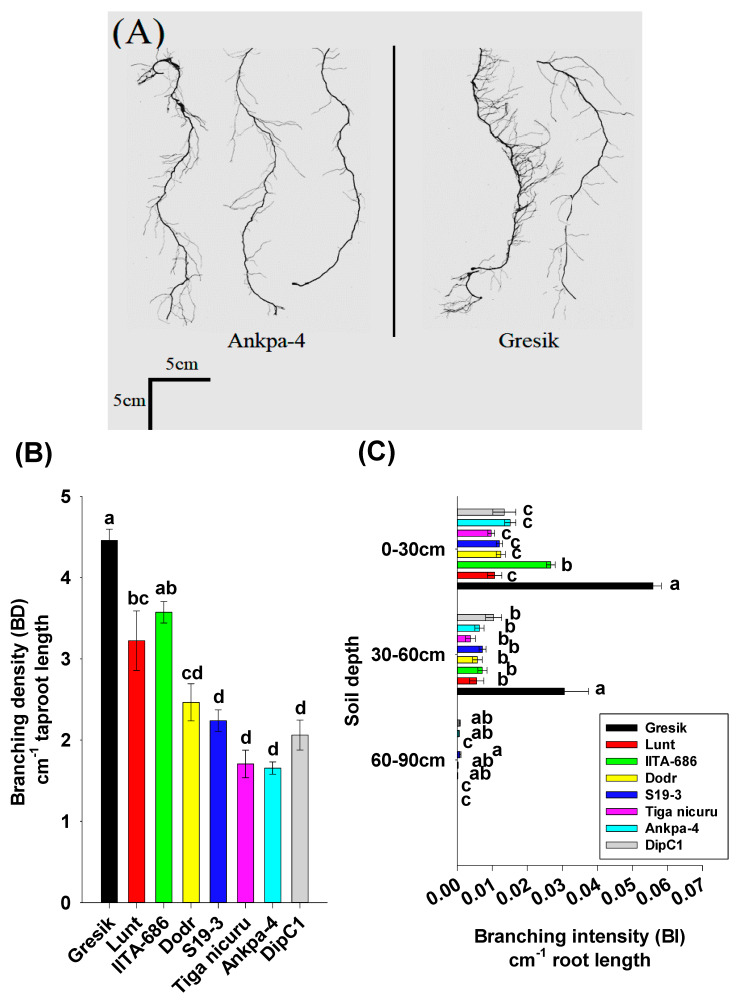
(**A**) Example bambara groundnut roots from different soil depths Ankpa-4 (from left to right 0 to 30, 30 to 60, and 60 to 90 cm) and Gresik (from left to right 0 to 30 and 30 to 60 cm) at 35 days after emergence (DAE) grown in a soil-filled PVC column of 20 × 110 cm (diameter and length, respectively); (**B**) Total branching density (BD) amongst different bambara groundnut genotypes; (**C**) Branching intensity (BI) amongst different bambara groundnut genotypes in different soil depth segments, i.e., 0 to 30, 30 to 60, and 60 to 90 cm. Mean ± se values (*n* = 6) are shown. Different letters indicate significant differences (HSD, *p* < 0.01).

**Figure 5 plants-09-00645-f005:**
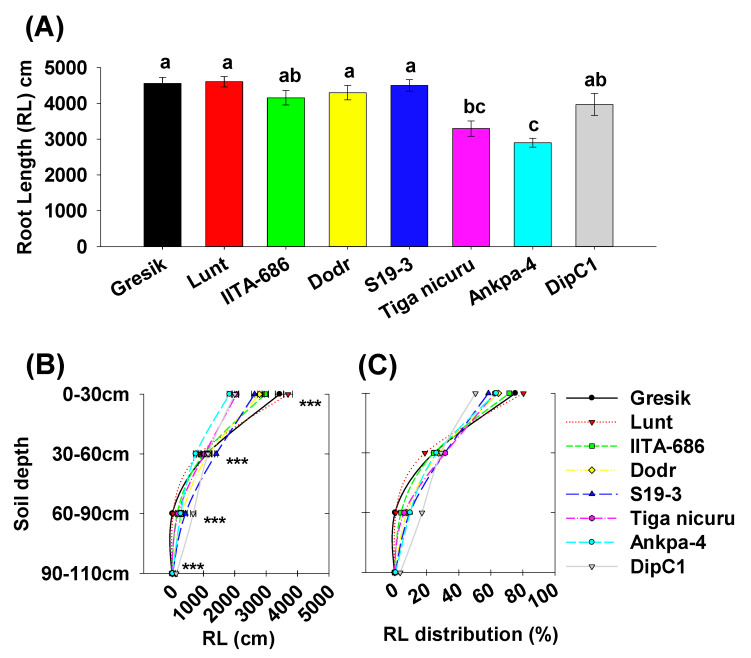
(**A**) Total root length (RL) (first- and second-order lateral roots). Mean ± se values (*n* = 12) are shown. Different letters indicate significant differences (HSD, *p* < 0.01); (**B**) RL’s per soil depth segments; (**C**) Average percentages of RL distribution per soil depth segment. Mean ± se values (*n* = 12) are shown. Significant differences as follows: * *p* < 0.05; ** *p* < 0.01; *** *p* < 0.001; and ns = not significant, amongst individual genotypes.

**Figure 6 plants-09-00645-f006:**
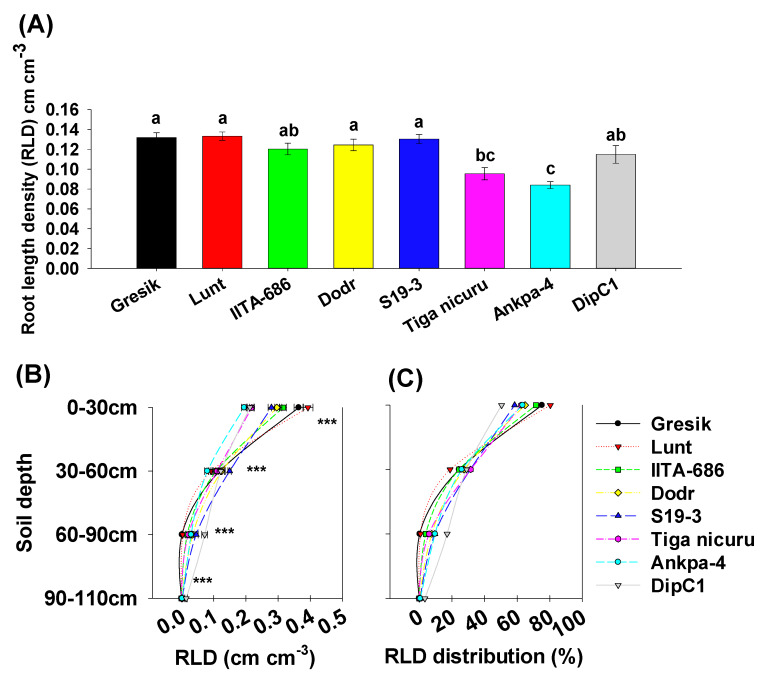
(**A**) Total root length density (RLD) (first- and second-order lateral roots). Mean ± se values (*n* = 12) are shown. Different letters indicate significant differences (HSD, *p* < 0.01); (**B**) RLD’s per soil depth segments; (**C**) Average percentages of RLD distribution per soil depth segment. Mean ± se values (*n* = 12) are shown. Significant differences as follows: * *p* < 0.05; ** *p* < 0.01; *** *p* < 0.001; and ns = not significant, amongst individual genotypes.

**Figure 7 plants-09-00645-f007:**
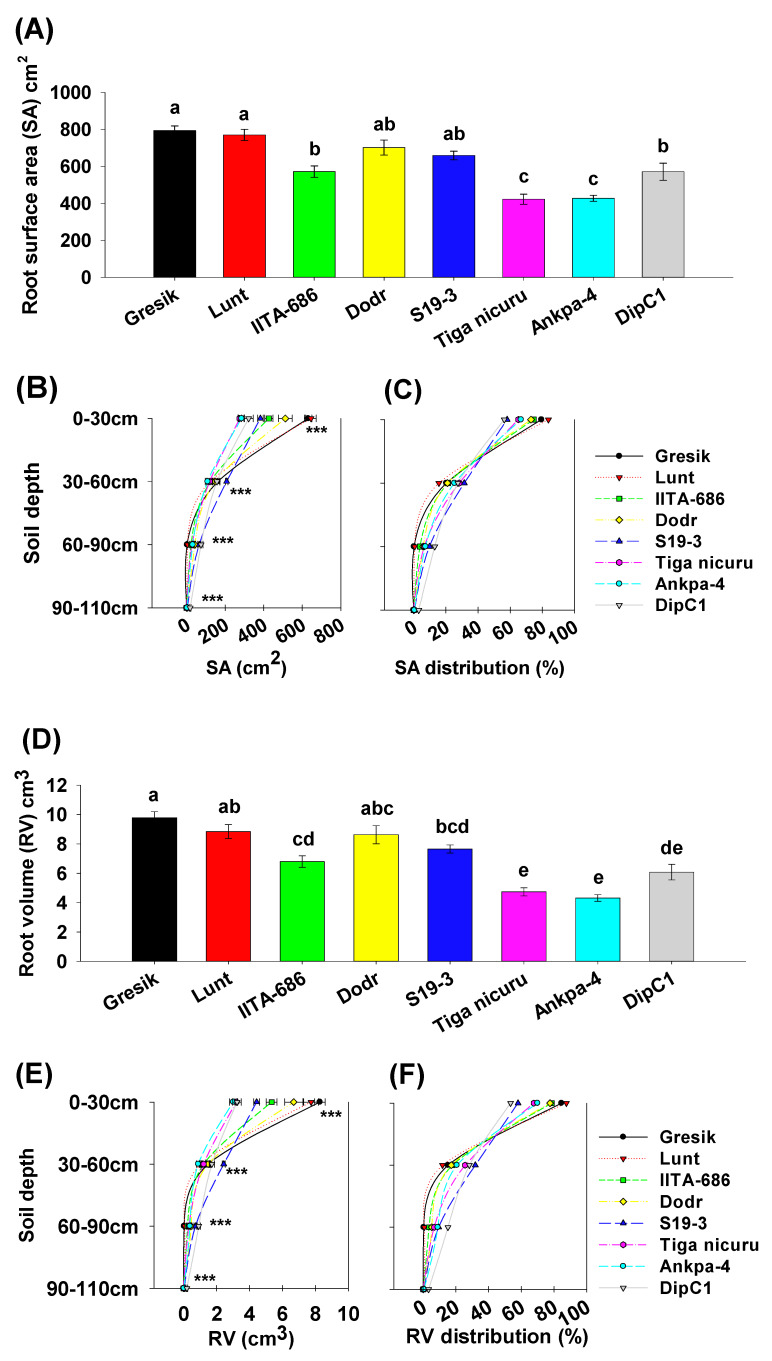
(**A**) Total root surface area (SA) in bambara groundnut genotypes. Mean ± se values (*n* = 12) are shown. Different letters indicate significant differences (HSD, *p* < 0.01); (**B**) SA per soil depth segments. Mean ± se values (*n* = 12) are shown. Significant differences as follows: * *p* < 0.05; ** *p* < 0.01; *** *p* < 0.001; and ns = not significant, amongst individual genotypes; and (**C**) Average percentages of SA distribution per soil depth segment; (**D**) Total root volume (RV) in bambara groundnut genotypes. Mean ± se values (*n* = 12) are shown. Different letters indicate significant differences (HSD, *p* < 0.01); (**E**) RV’s per soil depth segments. Mean ± se values (*n* = 12) are shown. Significant differences as follows: * *p* < 0.05; ** *p* < 0.01; *** *p* < 0.001; and ns = not significant, amongst individual genotypes; (**F**) Average percentages of RV distribution per soil depth segment; (**G**) Total root diameter (RDia) in bambara groundnut genotypes. Mean ± se values (*n* = 12) are shown. Different letters indicate significant differences (HSD, *p* < 0.01); (**H**) RDia per soil depth segments. Mean ± se values (*n* = 12) are shown. Significant differences as follows: * *p* < 0.05; ** *p* < 0.01; *** *p* < 0.001; and ns = not significant, amongst individual genotypes; and (**I**) Average percentages of RDia distribution per soil depth segment.

**Figure 8 plants-09-00645-f008:**
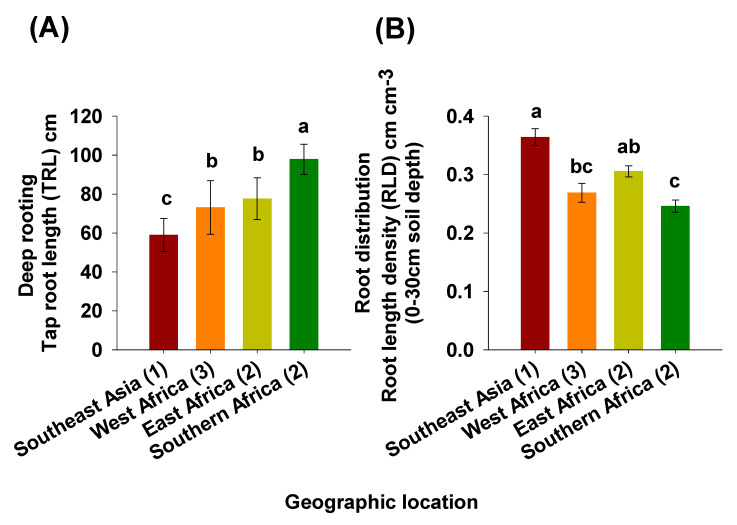
Graphical depiction of (**A**) mean deep rooting (taproot length in the 0 to 110 cm soil depth; TRL) and (**B**) root distribution (root length density in the 0 to 30 cm soil depth) bambara groundnut genotypes from the core parental line set originating from different geographical regions at 35 days after emergence (DAE) grown in a soil-filled PVC column of 20 × 110 cm (diameter and length, respectively). Different letters indicate significant differences (HSD, *p* < 0.01).

**Figure 9 plants-09-00645-f009:**
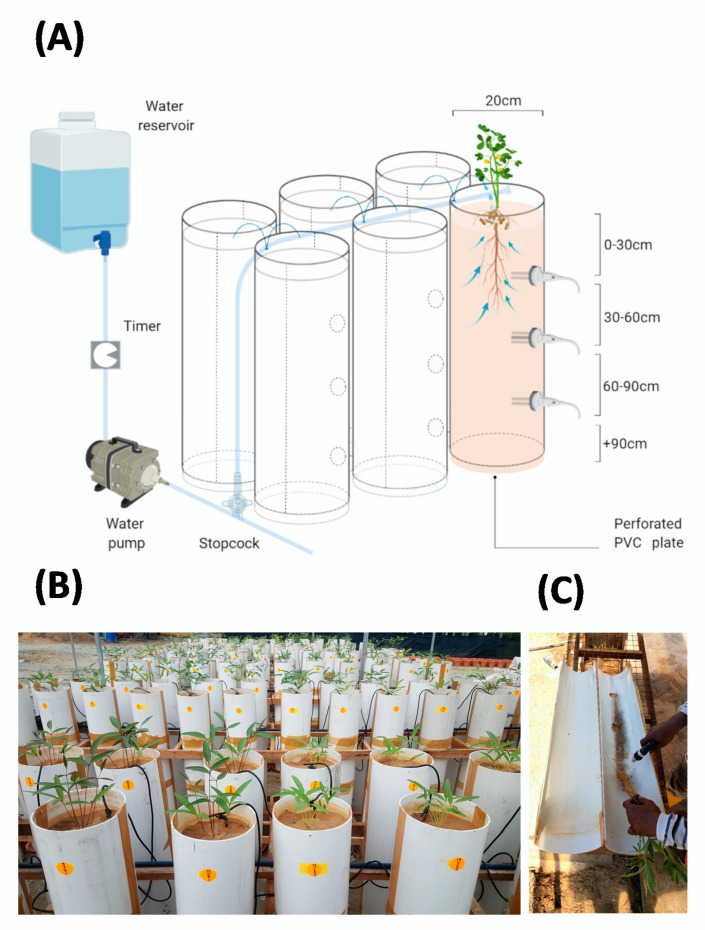
(**A**) Schematic representation of soil-filled PVC column of 20 × 110 cm (inside diameter and length, respectively), placed on a perforated plate. Control columns with probes installed to measure volumetric water content and soil temperature at 30, 60, and 90 cm depths. A 302 L reservoir tank with water or nutrient solution and a pump is used to feed the solution down irrigation pipes into the PVC columns. The pump is controlled by a timer, programmed for 15 min ON/OFF intervals. Figure created with BioRender.com; and (**B**) PVC column setup under rainout shelter including a wooden frame for structural support and (**C**) column split and root washing.

**Table 1 plants-09-00645-t001:** Effect of genotypes on days to 50% emergence (D50%) and shoot height (SH), number of leaves (NoL), and root to shoot (R:S) ratio at 35 days after emergence (DAE) grown in a soil-filled PVC column of 20 × 110 cm (diameter and length, respectively).

Genotypes	D50%	SH	NoL	R:S Ratio
	number of days	cm plant ^−1^	number plant ^−1^	RDW/SDW
Gresik	4.67 ± 1.03c	24.65 ± 1.1a	17.33 ± 1.91	0.35 ± 0.03a
Lunt	5.67 ± 1.03bc	23.02 ± 0.91ab	14.67 ± 0.95	0.19 ± 0.02bc
IITA-686	6.5 ± 1.05abc	24.1 ± 1.04ab	13.33 ± 0.8	0.21 ± 0.02bc
Dodr	5.33 ± 1.03bc	23.76 ± 0.96ab	13.83 ± 0.48	0.23 ± 0.02bc
S19-3	6.17 ± 1.47abc	22.7 ± 0.66ab	13.83 ± 0.54	0.28 ± 0.02ab
Tiga nicuru	5.5 ± 1.05bc	19.23 ± 2.15b	14.25 ± 1.09	0.23 ± 0.04bc
Ankpa-4	7.83 ± 1.47a	19.19 ± 1.52b	14.33 ± 0.71	0.16 ± 0.02c
DipC1	7.33 ± 1.03ab	23.08 ± 0.88ab	15.25 ± 1.42	0.21 ± 0.02bc
Mean	6.13	22.47	14.60	0.23
F probability				
Genotype	0.000 ***	0.009 **	0.27 ^ns^	0.000 ***

The data is mean ± se values (*n* = 12), except for NoL (*n* = 6), with different letters showing significant difference (HSD) as follows: * *p* < 0.05, ** *p* < 0.01, *** *p* < 0.001, and ns = not significant.

**Table 2 plants-09-00645-t002:** List of genotypes, respective seed color, and country collected, used for the soil-filled PVC column experiment.

Geographical Region	Genotypes ^1^	Seed Color	Country Collected	Climate	Rainfall Mean (mm year^−1^)
Southeast Asia	Gresik	Dark	Indonesia	Tropical wet	˃2000
West Africa	Lunt	Cream	Sierra Leone	Tropical wet	˃2000
	Ankpa-4	Brown	Nigeria	Tropical dry	˃2000
	Tiga nicuru	Cream	Mali	Subtropical	450
East Africa	IITA-686	Dark	Tanzania	Tropical dry	˃750
	Dodr	Red	Tanzania	Tropical dry	˃570
Southern Africa	S19-3	Dark	Namibia	Subtropical desert	350
	DipC1	Cream with black eye	Botswana	Semi-arid	500

^1^ Names mostly based on the place the seeds were collected, e.g., Gresik, city found in East Java, Indonesia; Lunt, Lungi the northern province of Sierra Leone; Ankpa, an area in Kogi State Nigeria; IITA-686, International Institute of Tropical Agriculture (IITA) Nigeria; Dodoma Red (Dodr), the national capital of Tanzania; and Diphiri Cream (DipC1), the region of Kweneng Botswana.

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
