# Peer review of "Root Foraging Capacity in Bambara Groundnut (Vigna Subterranea (L.) Verdc.) Core Parental Lines Depends on the Root System Architecture during the Pre-Flowering Stage"

_plants, 2020, doi:10.3390/plants9050645_

Round 1

Reviewer 1 Report

Line 36-37: “by means of” to “using”

Line 108 to 109, the figure 1 fonts need to be improved to be clearer especially on the legends of the x-axis.

Line 124: Figure 2 (a) looks very dark, is it possible to get a better picture?

Line 124: Figure 2 (b), the x-axis label fonts need to be separated by reducing the size

Line 163: Figure a, not clear enough

Line 163: figure (b) the x-axis label fonts need to be separated by reducing the size

Line 170: Figure (B) the x-axis label fonts need to be separated by reducing the size

Line 228: “clearly” is an unnecessary word

Line 233: after “In contrast” you need the comma after contrast

Line 480: “actually” is an unnecessary word

Line 522- 532: Figure S3, I am not sure why cluster 4 and one branch of cluster 3 was there when the 8 genotypes were already explained in cluster 1, 3, and 3. Three branches at the bottom don't have any name of the genotype to compare with and it seems unnecessarily there. The dendrogram can be cut to a linkage distance of 10.

Line 542-545; Table A1: can you include Genotype by environment interaction (GxE) in the analysis?

Line 542-545; Table A1: can you include H (heritability) for most of the traits?

Line 149: “amongst genotypes originating (form) drier versus wetter environments (Supplementary Figure S2”

amongst genotypes originating from drier versus wetter environments (Supplementary Figure S2

Author Response

Response to Reviewer 1 Comments

Point 1: Line 36-37: “by means of” to “using”

Response 1: Revised line 45 now reads as follows, -  In these contrasting habitats, bambara groundnut has diversified due to domestication from its wild relative, Vigna subterranea var. spontanea (Harms) Hepper, as a result of steady changes through natural and artificial selection [5].

Point 2: Line 108 to 109, the figure 1 fonts need to be improved to be clearer especially on the legends of the x-axis.

Response 2: We have made all figures text bold (line 113). In addition, to improve clarity and remove x-axis text overlap, we have adjusted the x-axis text to a 45-degree angle. This is a shift from our previous 30-degrees.

Point 3: Line 124: Figure 2 (a) looks very dark, is it possible to get a better picture?

Response 3: We have improved/brightened the image (line 130) by enhancing the brightness by 20% and reducing contrast by 20%.

Point 4: Line 163: Figure a, not clear enough

Response 4: We have improved/sharpened the low resolution image (line 170) by enhancing the overall sharpness by 10%.

Point 5: Line 163: figure (b) the x-axis label fonts need to be separated by reducing the size

Response 5: We have made all figures text bold (line 170). In addition, to improve clarity and remove x-axis text overlap, we have adjusted the x-axis text to a 45-degree angle. This is a shift from our previous 30-degrees.

Point 6: Line 170: Figure (B) the x-axis label fonts need to be separated by reducing the size

Response 6: We have made all figures text bold (line 178). In addition, to improve clarity and remove x-axis text overlap, we have adjusted the x-axis text to a 45-degree angle. This is a shift from our previous 30-degrees.

Point 7: Line 228: “clearly” is an unnecessary word

Response 7: Revised line 238 now reads as follows, - This indicated that high contrasting genotypes for the root traits studied could be distinguished and confirmed from the soil-filled PVC column system.

Point 8: Line 233: after “In contrast” you need the comma after contrast

Response 8: Revised line 246 now reads as follows, - In contrast, genotypes that originated from southern Africa i.e. DipC1 and S19-3 (both in cluster 4), had significantly deeper tap root systems than the other three regions (Fig. 8A) with also significantly less RLD in the shallow soil depth (Fig. 8B).

Point 9: Line 480: “actually” is an unnecessary word

Response 9: Revised line 499 now reads as follows, - This is especially valid for hot-dry-habitat S19-3 and DipC1, which flourish in an area of deep sandy soils under very dry and hot climate and in the case of our study, demonstrated the most noteworthy rooting traits.

Point 10: Line 522- 532: Figure S3, I am not sure why cluster 4 and one branch of cluster 3 was there when the 8 genotypes were already explained in cluster 1, 3, and 3. Three branches at the bottom don't have any name of the genotype to compare with and it seems unnecessarily there. The dendrogram can be cut to a linkage distance of 10.

Response 10: We use k-means clustering, with several starting values to determine the number of clusters (using our current linkage distance) that minimize the within-SS. This is discussed in section 4.5 Data analysis (line 488 to 491). Also we provide a detailed description (line 241 to 249) of the clustering results with clear reference to the respective genotypes used.

Point 11: Line 542-545; Table A1: can you include Genotype by environment interaction (GxE) in the analysis?

Response 11: Experiments were conducted in tandem during two consecutive seasons (i.e. 2017–2018 and 2018–2019). No variation between the two seasons for climatic data (humidity and temperature) were observed. This makes sense considering that these two years’ form part of the same season and therefore climatic conditions remained fairly constant with no statistically difference to warrant a G×E analysis. However, we choose to talk about G×Y (line 478) given that one out of the two experiments was carried out at the beginning of the new year. Similarly, Wald statistics revealed that the error components across the years G×Y, were also homogenous and therefore, we drew inferences combined across years. Non significant G×Y values were excluded - we felt this would be redundant considering our detailed explanation from line 476 – 479.

Point 12: Line 542-545; Table A1: can you include H (heritability) for most of the traits?

Response 12: In our experiment we use single genotypes (line 375) and as such estimates of variance components would only be at genotype level (as opposed to a more useful hybrid/crosses/population). So when dealing with single genotypes in a single environment, we are only getting an idea of likelihood and this can be misleading especially if the sample size and in our case environment sites/treatments are limited.

Point 13: Line 149: “amongst genotypes originating (form) drier versus wetter environments (Supplementary Figure S2” amongst genotypes originating from drier versus wetter environments (Supplementary Figure S2)

Response 13: Revised line 166 now reads as follows, - On the other hand, shoot height was closely and positively correlated (P<0.05) with branching number in the deep 60-90cm of the soil (R2=0.53), and this was largely amongst genotypes originating from drier versus wetter environments (Supplementary Figure S2).

Reviewer 2 Report

Dear Authors,

please find in attached the pdf, with underline some part that should need some revisions. However, in the complex, the article satisfies me. 

Best Regards,

FG

Author Response

Response to Reviewer 2 Comments

Point 1: From personal experience, in a situation of combined stress such as drought and phosphate deprivation, the Water Use Efficiency (WUE) was higher than the two single stress. Probably I miss your point, try to rewrite differently, or check again what are you arguing.

Response 1: Revised line 309 now reads as follows, - From a functional perspective, a costly highly branching system in the shallow top soil layer improves the root absorption of phosphorous, however in case of a drought, this would enhance water depletion in that layer because of acute root competition [50].

Point 2: Please use SI. Convert inch to cm

Response 2: Revised line 402 now reads as follows, - To facilitate root harvesting, each column was cut longitudinally along both sides and the two halves taped together with 4.8cm (by width) packing brown tape before filling with soil.

Point 3: Line 396: (10kg ha−1 N as urea (46%), 50kg ha−1 P as Christmas island rock phosphate (CIRP) (30%) and 50kg ha−1 K as muriate of potash (MOP) (60%).

Usually, in a greenhouse experiment with pots, the fertiliser quantity is indicated in Molarity than the Kg per hectare. So, please modify it in Molarity (M or mM) and then you can indicate the percentage

Response 3:  We are more interested in the field equivalence and I would not agree that molarity is used except in non-soil experiments

Point 4: Line 400: The solution consisted of Mn EDTA (3.8%), Fe EDTA (4.0%), Cu EDTA (1.5%), B (0.5%), Zn EDTA (1.5%), Co (0.03%), Mo (0.10%), and Mg (5.10%).

Usually, in a greenhouse experiment with pots, the fertiliser quantity is indicated in Molarity than the Kg per hectare. So, please modify it in Molarity (M or mM) and then you can indicate the percentage

Response 4:  We are more interested in the field equivalence and I would not agree that molarity is used except in non-soil experiments

Point 5: How long did you treat the seeds with Clorox?

Response 5: Revised line 421 now reads as follows, - After removing broken and damaged seeds, uniform sized seeds were selected and surface sterilised in a 10% (v/v) Clorox solution (sodium hypochlorite 0.5%) for 2 min on a rotary shaker at 150 rpm.

Point 6: It took a while to understand this protocol. Please, try to rewrite in a way that's less difficult

Response 6: Revised line 446 to 466 now reads as follows, - To extract the roots, the PVC column was laid down and tilted at a 20° angle to the root washing station. The detachable perforated PVC plate was removed and the column split in half longitudinally. The soil was gradually removed to expose the roots in bottom-up manner using soft spray watering head (Fig 1B). After complete removal of the soil, the shoots (i.e. leaves and stems) were separated from roots and entire root systems were submerged in water-filled zip lock bags of 22cm×30cm (width and length respectively) and transported to the laboratory for further assessment. In order to identify and measure the tap root length (TRL), entire roots were laid flat and stretched against a two-meter ruler, giving an estimate of the deepest extent of the root system. For the purposes of this paper, entire root systems (i.e. totals) were analysed first. Following totals analysis, root systems were cut into different segments with respect to varying 30cm soil depth (i.e. 0-30, 30-60, 60-90 and 90-110cm) and analysed as such. In both cases, roots were spread in a shallow A3 size – 297mm×420mm (height×width respectively) clear acrylic tray filled with water and disentangled using plastic forceps to reduce overlapping. Root traits were all computed from the scanned images in greyscale at 400 dots per inch using a flatbed Epson Scanner (Epson Perfection V700, CA, USA) with WinRhizo Pro software v2009 (Regent Instruments, Montreal, QC, Canada). These included root length (RL cm), representing root lengths in the network. Branching number (BN), the number of first-order lateral roots emerged from the tap root. Root surface area (SA cm2), root volume (RV cm3) and root diameter (RDia mm) assessed as proportionate estimations of RL and expected to exhibit the same pattern and trend of variation [45]. These traits subsequently allowed for the calculation of root length density (RLD cm cm3), branching density (BD) and branching intensity (BI) using the following formula’s: